# Effect of N-Acetyl-L-cysteine on Activity of Doxycycline against Biofilm-Forming Bacterial Strains

**DOI:** 10.3390/antibiotics12071187

**Published:** 2023-07-14

**Authors:** Tsvetelina Petkova, Nikolina Rusenova, Svetla Danova, Aneliya Milanova

**Affiliations:** 1Department of Pharmacology, Animal Physiology, Biochemistry and Chemistry, Faculty of Veterinary Medicine, Trakia University, 6000 Stara Zagora, Bulgaria; ts_petkova87@abv.bg; 2Department of Veterinary Microbiology, Infectious and Parasitic Diseases, Faculty of Veterinary Medicine, Trakia University, 6000 Stara Zagora, Bulgaria; n_v_n_v@abv.bg; 3The Stephan Angeloff Institute of Microbiology, BAS, 26 Georgi Bonchev Str., 1113 Sofia, Bulgaria; stdanova@abv.bg

**Keywords:** *Staphylococcus aureus*, *Escherichia coli*, *Pseudomonas aeruginosa*, doxycycline, N-acetyl-L-cysteine, biofilm

## Abstract

Biofilm-forming bacteria are associated with difficult-to-cure bacterial infections in veterinary patients. According to previous studies, N-acetyl-L-cysteine (NAC) showed an inhibitory effect on biofilm formation when it was applied in combination with beta-lactam antibiotics and fluoroquinolones. The lack of information about the effect of NAC on doxycycline activity against biofilm-forming strains was the reason for conducting this study. *Staphylococcus aureus* (*S. aureus*) ATCC 25923, *Staphylococcus aureus* O74, *Escherichia coli* (*E. coli*) ATCC 25922 and *Pseudomonas aeruginosa* (*P. aeruginosa*) ATCC 27853 were used to evaluate the activity of doxycycline with and without addition of NAC on planktonic bacteria and on biofilm formation. The minimum inhibitory concentrations (MICs) of doxycycline were not affected by NAC for Gram-negative strains and were found to be two times higher for the strains of *S. aureus*. The minimum biofilm inhibitory concentrations (MBICs) for Gram-negative bacteria (2 μg/mL for *E. coli* ATCC 25922 and 32 μg/mL for *P. aeruginosa* ATCC 27853), determined using a standard safranin colorimetric assay, were higher than the MICs (0.5 and 4 μg/mL, respectively). The data suggest that the combinations of doxycycline and NAC could stimulate the growth of planktonic cells of *S. aureus* and biofilm-forming *E. coli* ATCC 25922. NAC did not affect the strong inhibitory effect of doxycycline on the biofilm formation by the strains of *S. aureus.*

## 1. Introduction

*Staphylococcus aureus* (*S. aureus*), *Escherichia coli* (*E. coli*) and *Pseudomonas aeruginosa* (*P. aeruginosa*) are among the pathogens often associated with severe and chronic infections in both animals and humans. Several of them are able to produce biofilm and cause infections that are difficult to treat [1,2,3]. Biofilm forming is defined as the ability of bacteria to aggregate and produce extracellular matrix, which protects them, increases their survival rate, and allows attachment to abiotic and biotic surfaces [4,5]. Biofilm-producing strains exhibit increased resistance to antimicrobial agents, which can be explained by the limited access of antibiotics and cells of the immune system to the bacteria [1,6,7]. The decreased susceptibility of these strains to antibacterial drugs can be related to genetically obtained antibiotic resistance (AR) or can be defined as phenotypic resistance, also called tolerance [8,9]. Increasing problems with antimicrobial resistance and the need to gain a better understanding of the infections associated with biofilm-producing microorganisms triggered efforts to find new solutions to combat biofilm-associated resistance. The combination of antibiotics with drugs that can disrupt biofilm formation are among the innovative strategies for treating biofilm-related infections.

N-acetyl-L-cysteine (NAC) is a widely used drug in human and veterinary clinical practices. It has been applied in the treatment of acetaminophen toxicity [10], otitis [11], corneal ulcers [12], infections of the oral cavity [13] and as a mucolytic agent in the treatment of lower respiratory tract diseases or endometritis and endometriosis [14,15]. It also has antioxidant, cytoprotective and anti-inflammatory properties [16,17]. In addition, the published literature shows that NAC has an adverse effect on biofilm formation and decreases its formation when combined with penicillins, tetracyclines, fluoroquinolones and aminoglycosides [13,18,19,20]. These interactions are not always synergistic, and some of them are classified as indifferent or antagonistic depending on the pathogenic strain and the antibacterial drug used.

Tetracyclines are among the most often applied antibiotics in almost all animal species [21]. According to the last categorization of antibiotics for veterinary medicine, suggested by European Medicines Agency (EMA), they should be used as first-line treatments if possible [22]. Therefore, finding the most successful strategy for the prudent and responsible use of tetracyclines requires more investigations on the possibility for a reduction in cases of treatment failure and in the risk of resistance selection. The combination of antibacterial agents and substances with antibiofilm properties can induce drug synergism and can increase the sensitivity of biofilm-producing bacteria, restoring the susceptibility of the pathogens [18]. Achieving such a positive cumulative effect, especially with tetracyclines (known to have transferable induced AR), remains an important necessity. The scarcity of data and their controversial nature on the effect of combinations of tetracyclines and NAC on various biofilm-producing microorganisms was the motivation to perform the present in vitro experiments. This investigation aimed to evaluate the antibacterial activity of doxycycline and NAC, used either alone or in combination, towards the biofilm-producing strains *Staphylococcus aureus*, *Pseudomonas aeruginosa* and *Escherichia coli*.

## 2. Results

### 2.1. Determination of the Minimum Inhibitory Concentrations (MICs) of Doxycycline Hyclate and N-Acetyl-L-cysteine

Four bacterial strains belonging to pathogenic species were studied in a model microplate system. During the cultivation in MHB medium, they were incubated with doxycycline hyclate and with NAC. The MICs for doxycycline and NAC were determined at twofold serial dilutions in the range of 128–0.0625 µg/mL and 8000–0.122 µg/mL, respectively. Doxycycline, alone and in combination with NAC, showed a broad spectrum of activity against the tested *E. coli*, *S. aureus* and *P. aeruginosa*, and strong inhibition with specific MIC concentrations (Table 1).

The MIC values of doxycycline were 2- to 16-fold higher against the investigated Gram-negative strains in comparison to the Gram-positive ones (Table 1), while a similar effect of the antibiotic on the growth of both strains, *S. aureus* O74 and *S. aureus* ATCC 25923, must be noted. At the same time, NAC showed an equal value of MIC for all the tested strains, and it was higher than that of doxycycline. For the combination of antibiotic (tested at the range of 128–0.0625 µg/mL) with NAC (fixed concentration of 1 µg/mL), the values of MIC remained unchanged for *E. coli* ATCC 25922 and *P. aeruginosa* ATCC 27853. The combination of both drugs resulted in fourfold higher values of MIC for *S. aureus* ATCC 25923 and *S. aureus* O74 (Table 1).

In the presence of a sub-inhibitory concentration of doxycycline, moderate-to-strong inhibition of the pathogens was observed after 24 h cultivation (Figure 1c,d). A broad spectrum of effects, from the inhibition (Figure 1c) to the stimulation (Figure 1a,b,d) of bacterial growth, when the combination of doxycycline with NAC was applied must be noted.

### 2.2. Biofilm Formation of Gram-Positive and Gram-Negative Pathogens

The safranin staining protocol was used to assess biofilm formation by the tested strains after cultivation for 48 h at 37 °C. Since all the investigated strains were classified as moderate (the strain of *E. coli*) to strong (the strains of *S. aureus* and *P. aeruginosa*) biofilm producers, they were used in the in vitro investigations of antibiofilm activity of doxycycline and NAC (Figure 2).

### 2.3. The Effect of Doxycycline and N-Acetyl-L-cysteine on Biofilm Formation

Doxycycline, at concentrations between 0.125 and 32 μg/mL, inhibited the biofilm formation of all the tested strains (Table 2).

The co-administration of NAC, added at a concentration of 1 μg/mL, did not change the MBIC of doxycycline for most of the tested strains, with the exception of *E. coli* ATCC 25922. The combination of doxycycline with NAC caused a two-fold increase in the value of MBIC for *E. coli* ATCC 25922 (Table 2). A tendency to stimulate the growth of *E. coli* ATCC 25922 was observed at concentrations of doxycycline ≤2 μg/mL and in the presence of NAC (Figure 3c). Low values of MBIC were observed for both strains of *S. aureus* after the application of doxycycline, alone or in combination with NAC (1 μg/mL) (Figure 3a,b). The effect of doxycycline on the biofilm growth of *P. aeruginosa* ATCC 27853 was not consistent (Figure 3d). A concentration of doxycycline ≥32 μg/mL was able to cause 90% inhibition of the growth of *P. aeruginosa* ATCC 27853. The addition of NAC did not change the activity of doxycycline.

NAC, administered alone, significantly inhibited the growth of *S. aureus* ATCC 25923 and *E. coli* ATCC 25922 at concentrations ≥ 2000 µg/mL. The biofilm density was reduced by >90% for these strains as well as for *S. aureus* O74 at levels of NAC ≥4000 µg/mL. The inhibition of the growth of *S. aureus* O74 was inconsistent, although a significant reduction in the biofilm was observed at concentrations of 0.98–1.95 and between 62.5 and 8000 µg/mL. The administration of NAC at a concentration of 4000 µg/mL resulted in 90% inhibition of the growth of *P. aeruginosa* ATCC 27853, although it was significantly inhibited at the level of 2000 µg/mL (Table 2).

## 3. Discussion

Reducing the risk for the selection of resistant pathogens requires the precise and prudent use of antibiotics and urgent measures for finding effective dosage regimens of existing antibacterial agents to combat biofilm-forming pathogens. The tolerance and resistance to antimicrobial agents of pathogenic strains of *S. aureus*, *E. coli* and *P. aeruginosa* are often associated with their ability to form biofilms. Therefore, in the present work, a combination of doxycycline and NAC, widely used in clinical and veterinary practice, was assessed in vitro. With this aim, strains of *E. coli*, *S. aureus* and a highly biofilm-forming strain of *P. aeroginosa* were selected for the study.

Sub-inhibitory concentrations of the antibiotics doxycycline and tetracycline can have biofilm-enhancing effects against *E. coli*, or can be almost inefficient against biofilm-embedded *S. aureus* and *P. aeruginosa* [23,24]. Additionally, protein binding can decrease the antimicrobial activity of these antibiotics in vivo [25]. Data from the literature show that the protein binding of tetracyclines is variable and 50 to 93% binding has been reported for doxycycline, depending on the drug concentration and bird or mammalian species [26,27]. Therefore, finding a drug that can increase the activity of doxycycline against biofilm-forming pathogens can have clinical significance. NAC has been widely discussed as a drug with a putative role in preventing biofilms or in inhibitory effects at various stages of its formation [18,19,20]. NAC can modulate the antibacterial activity of antibiotics from different groups, including tetracyclines [28,29]. The NAC–antibiotic combination may result in synergistic, neutral or antagonistic antibacterial activity. To the best of our knowledge, the data from the current experiment show different effects of the combination of doxycycline and NAC for the first time. Furthermore, these effects could not be predicted because they are strain-dependent and bacterial cells under biofilm may respond differently to antibiotics if compared to planktonic culture [7].

Initially, the values of MIC were determined in planktonic cultures of the investigated four strains after 24 h cultivation. Both strains, *S. aureus* ATCC 25923 and *S. aureus* O74, were sensitive to doxycycline, with a value of MIC lower than the breakpoint of 1 μg/mL [30]. Our data are in accordance with the results by Jones and Stilwell [31]. Methicillin-susceptible strains of *S. aureus* showed MIC_90_ 0.12 μg/mL and were four-fold more susceptible to doxycycline in comparison to methicillin-resistant *S. aureus*, with MIC_90_ 0.5 μg/mL [31]. The MIC values of doxycycline were four-fold higher when it was applied in combination with NAC, added at concentrations of 1 μg/mL. These concentrations are achievable in the bodies of chickens in steady-state conditions after administration at a dose rate of 100 mg/kg body weight for five consecutive days [32]. Although both investigated *S. aureus* strains remained susceptible to doxycycline, with MICs equal to 1 μg/mL, the administration of doxycycline with NAC can decrease the efficacy of the antibiotics against these Gram-positive strains, which could be a reason for therapeutic failure. One of the limitations of our study is that it does not allow drawing a strong conclusion for clinical practice due to the limited number of tested strains and the absence of in vivo investigations. The MIC value of doxycycline for *E. coli* ATCC 25922 was below the CLSI breakpoint of 4 μg/mL and the addition of NAC did not change the activity of this tetracycline antibiotic [33,34]. These results should be discussed with caution because a lack of susceptibility to tetracyclines among *E. coli* clinical isolates was determined, and it is difficult to predict the impact of NAC on doxycycline activity if more strains are investigated [35]. The other investigated Gram-negative bacterial strain, *P. aeruginosa* ATCC 27853, was sensitive to doxycycline, with or without the addition of NAC, which is in line with the reported data for strains of *P. aeruginosa* [36,37]. Wide variability in the susceptibility of *Pseudomonas* isolates to tetracycline was noticed, which suggests that further investigations of doxycycline–NAC interactions are required [37]. It can be suggested that the activity of tetracycline antibiotics for Gram-negative bacteria was not affected by the presence of NAC because it altered neither the MICs of doxycycline for the investigated strains of *E. coli* and *P. aeruginosa* nor the activity of tetracycline against strains of the same bacterial species in previously reported experiments [28]. Our data are similar to the published findings in the lack of effect of NAC on the antibacterial activity of tetracycline for Gram-negative *Prevotella intermedia* [13].

The antimicrobial properties of NAC against pathogenic bacteria were described in previous studies; therefore, its antimicrobial activity was tested with the four strains included in our experiments in planktonic cultures [13,19,20]. They were partly explained by the acidifying effect of NAC. Confirming the results from other studies, we found high MIC values of NAC (4000 µg/mL) for the investigated strains of *S. aureus*, *E. coli* and *P. aeruginosa* [12]. The cited authors reported MIC values of NAC equal to 3120 µg/mL for clinical isolates of *P. aeruginosa* and between 1560 and 3120 µg/mL for *S. pseudintermedius* [12]. The MIC values of clinical isolates of *P. aeruginosa* from dogs with otitis externa were between 2500 and 10,000 µg/mL [11]. Such high concentrations cannot be achieved or maintained in vivo. Pharmacokinetic studies showed that the maximum plasma concentrations after the intravenous or oral administration of NAC, at a dose rate of 100 mg/kg b.w. in cats or after its oral administration at the same dose in chickens, were not higher than 394.30 ± 71.44, 19.66 ± 6.85 and 2.26 ± 0.91 µg/mL, respectively [10,32]. In another experimental design with *Prevotella intermedia*, concentrations of NAC between 375 and 750 µg/mL showed the ability of this thiol-reducing agent to lower the redox potential of the culture medium, resulting in the stimulation of bacterial growth [13]. These data indicate that NAC’s antibacterial activity can be expected at very high concentrations after topical administration. Although NAC is often combined with antibiotics, more investigations on its potential synergistic or antagonist effect are necessary before its administration for the topical treatment of bacterial infections.

As the second step, the effect of NAC on the activity of doxycycline in biofilm formation by *S. aureus* ATCC 25923, *S. aureus* O74, *E. coli* ATCC 25922 and *P. aeruginosa* ATCC 27853 was determined. All of the strains included in our experiments were biofilm producers, which allowed performing the tests for biofilm formation. Doxycycline showed a strong inhibitory effect on the biofilm formation by the strains of *S. aureus*. Our results are in line with previous findings for the high activity of 0.5 to 50 µg/mL doxycycline against biofilm formation by *S. aureus* ATCC 25923 [38]. In contrast to the observations for Gram-positive bacteria, higher MBICs were determined for Gram-negative bacteria *E. coli* ATCC 25922 and *P. aeruginosa* ATCC 27853. The biofilm viability of another Gram-negative bacterium, *Prevotella intermedia*, was significantly reduced by >20 µg/mL tetracycline, which was similar to our findings [13]. The results of our tests revealed a lack of a statistically significant effect of NAC (1 µg/mL) on the activity of doxycycline in the prevention of biofilm formation by the investigated strains of *S. aureus* and *P. aeruginosa*. NAC showed different effects on the biofilm formation capacity of the tested strain *E. coli* ATCC 25922 when it was applied in combination with sub-inhibitory concentrations of doxycycline, lower than the MBIC for the antibiotic. A similar tendency to stimulate biofilm was observed in a previous study, when higher concentrations of NAC, between 750 and 3000 µg/mL, were applied [13]. The antibiofilm effect of tetracycline for Gram-negative bacteria *Prevotella intermedia* was slightly decreased by NAC, which was partly explained by the lowering of the redox potential of the culture medium [13].

The investigations on the antibacterial activity of NAC against biofilm-forming Gram-negative and Gram-positive strains were performed on the basis of the available information about its antibiofilm activity [39]. Several mechanisms behind this action have been proposed: the modulation of bacterial cell adherence and the prevention of biofilm formation, impairment of the matrix architecture, and the stimulation of biofilm disruption [39]. Our data showed that the MBICs of NAC were as high as the values of MIC for planktonic cultures of the investigated strains. In vitro studies revealed that NAC was able to reduce biofilm formation in cultures of *S. epidermidis* at levels >250 µg/mL [40]. According to Zhao and Liu [41], NAC was able to detach the mature biofilm of *P. aeruginosa* PAO1 at concentrations of 500 µg/mL and succeeded in completely disrupting it at 10,000 µg/mL. Biofilm formation by *P. aeruginosa* and uropathogenic *E. coli* strains was significantly decreased by NAC at concentrations of 500 µg/mL, 4- to 8-fold lower than those observed in our study [42]. NAC, as an acidic compound, applied at high concentrations, was able to penetrate the biofilm and bacterial membrane and increase the intracellular oxidative status, in this way killing the bacterial cells [43]. Altogether, these data suggest that NAC at concentrations achievable in vivo was not able to modulate the antibiofilm activity of doxycycline and can inhibit biofilm formation at high concentrations in our experimental conditions (≥4000 µg/mL). Based on these results, only the topical administration of NAC can be recommended if achieving antibiofilm activity is required. The effect of the combination of doxycycline and NAC on biofilm formation can be defined as indifferent for *S. aureus* and *P. aeruginosa*. The co-administration of these drugs against *E. coli* showed potential for the stimulation of biofilm growth when the antibiotic was at sub-inhibitory concentrations, which could be unfavorable from a clinical point of view. More investigations are required in order to evaluate the possible synergistic combinations of antibacterial drugs and NAC and to exclude their undesirable interactions.

## 4. Materials and Methods

### 4.1. Bacterial Strains, Culture Media and Conditions

Four pathogenic strains were included in the study. *Staphylococcus aureus* American Type Culture Collection (ATCC) 25923, *Escherichia coli* ATCC 25922 and *Pseudomonas aeruginosa* ATCC 27853 were obtained from the Bulgarian National Collection for Microorganisms and Cell Cultures (NBIMCC, Sofia, Bulgaria). *Staphylococcus aureus O74* was isolated from a mastitic cow and was kindly provided by Utrecht University, the Netherlands. All tested microorganisms were stored at −20 °C according to the provider’s instructions, or in respective media supplemented with 20% *v/v* glycerol. Prior to the experiments they were restored in Tryptone Soya Broth (TSB, HiMedia Laboratories GmbH, Einhausen, Germany) and pre-cultivated twice before use in Tryptic Soya Agar (TSA, Sigma-Aldrich, Darmstadt, Germany) supplemented with 5% *v/v* defibrinated sheep blood at 35 °C for 20–24 h.

Microbiological tests were performed by using cation-adjusted Mueller–Hinton broth (MHB, HiMedia Laboratories GmbH, Einhausen, Germany), TSB and TSA, prepared and sterilized according to the providers’ instructions. Bacterial inoculum for minimum inhibitory concentrations and the other in vitro tests were prepared using the direct single-colony suspension method in physiological saline. Approximately 1.5 × 10^8^ cfu/mL was serially diluted to achieve a final concentration of 5 × 10^5^ cfu/mL, according to guidance that corresponded to the 0.5 McFarland standard, determined with Densilameter II (Erba Lachema, Brno, Czech Republic) [33].

### 4.2. Drugs and Reagents

N-acetyl-L-cysteine (NAC, TLC, ≥99%, Sigma Aldrich, St. Louis, MO, USA), doxycycline hyclate (HPLC grade ≥ 98%, Sigma Aldrich, St. Louis, MO, USA) and safranin for microbiology (Safranin, Chimtex OOD, Dimitrovgrad, Bulgaria) were used in the current investigation. Hydrochloric acid (ACS reagent, 37%, Sigma Aldrich, St. Louis, MO, USA) was used as 0.1 M solution. Sodium hydroxide (ACS reagent, ≥97.0%, Sigma Aldrich, St. Louis, MO, USA) was used as 0.2 M solution. The antibiotic and NAC were dissolved in the corresponding cultural media (TSB or MHB).

### 4.3. The Determination of Minimum Inhibitory Concentrations (MICs)

The minimum inhibitory concentrations (MICs) of doxycycline hyclate and NAC against the tested strains were determined using the standard broth microdilution method, following the Clinical and Laboratory Standards Institute guidelines [33]. Stock solutions of doxycycline hyclate (1 mg/mL) and NAC (16 mg/mL) were prepared in Mueller–Hinton broth (MHB) on the day of the experiment. Serial two-fold dilutions of both compounds were prepared. The antibacterial activity of doxycycline was tested at concentrations from 128 to 0.0625 µg/mL, and of NAC from 8000 µg/mL to 0.12 µg/mL. The MIC values of doxycycline at concentrations ranging from 128 to 0.0625 µg/mL in combination with 1 µg/mL NAC were also determined. The concentration range of the drugs was selected according to the results from preliminary tests and to the literature data [44,45,46]. MHB medium was used as a negative control. The tested bacterial strains, cultivated in MHB without the addition of drugs, served as a positive control. The tests were performed in 96-well flat-bottom plates (Costar, Corning Incorporated, Kennebunk, ME, USA). The samples were cultivated at 37 °C for 24 h. Thereafter, the optical density (OD) was measured at a wavelength of 620 nm (Synergy LX Multi-Mode Microplate Reader, BioTek, Winooski, VT, USA). The MIC was defined as the lowest drug concentration resulting in an OD value close to blank. The independent experiments were performed in triplicate.

### 4.4. In Vitro Biofilm Formation and the Determination of Minimum Biofilm Inhibitory Concentration (MBIC)

Biofilm formation was assessed by applying the standard safranin colorimetric assay as described earlier [47]. The same method was applied to determine the MBIC of doxycycline and NAC, alone and in combination. Briefly, the stock solutions of doxycycline hyclate (256 µg/mL) and NAC (16,000 µg/mL) were prepared in TSB on the day of the experiment. Serial dilutions of the drugs were placed in U-bottom 96-well plates (Costar, Corning Incorporated, Kennebunk, ME, USA) in triplicate. An aliquot of 100 μL of TSB with the appropriate drug concentration was placed in the wells; then, 100 μL of the bacterial suspension of the respective strain (1 × 10^7^ cfu/mL) was added to reach final doxycycline concentrations of 128, 64, 32, 16, 8, 4, 2, 1, 0.5, 0.25, 0.125 and 0.0625 µg/mL. The tested concentrations of NAC were 8000, 4000, 2000, 1000, 500, 250, 125, 62.5, 31.2,15.6, 7.8, 3.9, 1.95, 0.975, 0.488, 0.244 and 0.122 µg/mL. The final concentration of the bacterial suspension was 5 × 10^6^ cfu/mL. Additionally, the effects of the combination of doxycycline with NAC on the biofilm formation capacity of the investigated strains were tested at tetracycline antibiotic concentrations between 128 and 0.0625 µg/mL and 1 µg/mL NAC. The concentration of NAC was selected on the basis of previously performed pharmacokinetic studies with broiler chickens, orally treated with a dose of 100 mg/kg b.w. [32]. Wells with 200 μL of sterile TSB (*n* = 3 replicates) were used as a negative control for biofilm formation, and wells containing 200 μL of TSB with 5 × 10^6^ cfu/mL bacterial suspension of the respective strain served as a positive control (*n* = 3 replicates). The plates were covered with sterile, gas-permeable breathseal sealer (Greiner Bio-One GmbH, Frickenhausen, Germany). They were incubated at 37 °C for 24 h with constant shaking at 150 rpm (Heidolph Shakers & Mixers–Temperature Controlled Shaking, Schwabach, Germany) and another 24 h at 37 °C without shaking. Thereafter, the culture medium of each well was discarded and 200 µL of 0.1 M hydrochloric acid was added to the wells. The plates were left at room temperature for 1 h; then, the hydrochloric acid was removed from the wells. The plates were stained with 200 µL of 0.1% *w/v* solution of safranin in every well and they were left for 45 min at room temperature. Non-bound safranin was removed by rinsing the wells three times with deionized water before being allowed to dry. The dye bound to the adherent cells was solubilized in 150 µL of 0.2 M sodium hydroxide solution for 60 min at 57 °C with constant shaking at 150 rpm. The contents of the plates were transferred to flat-bottom plates (Costar, Corning Incorporated, Kennebunk, ME, USA) and the optical density was measured at 540 nm (Synergy LX Multi-Mode Microplate Reader, BioTek Instruments, Inc., Winooski, VT, USA). All tests were performed in triplicate.

The investigated pathogens were determined as biofilm producers if the OD ± SD of the sample with each bacterial strain tested in triplicate was ≥2 OD of that of the negative control (the wells only with TSB), according to Stepanovic et al.’s method [48]. The strains with measured values between 2 × OD and 4 × OD were defined as weak biofilm producers, while those with values ≥4 × OD of TSB were defined as strong ones [49]. The minimum biofilm inhibitory concentration (MBIC) was defined as the lowest amount of drugs that caused at least 90% inhibition of biofilm biomass compared to the untreated positive control, according to the following equation:% biofilm inhibition = [1 − (OD test/OD positive control)] × 100,(1)

### 4.5. Statistical Analysis

All data are presented as mean ± standard deviation (SD). A statistical evaluation of the biofilm assay results was performed using one-way ANOVA (Statistica for Windows 10.0, StatSoft, Inc., Tulsa, OK, USA). A *t*-test was applied for post hoc analysis. Differences were considered statistically significant at *p* < 0.05.

## 5. Conclusions

In conclusion, our data suggest that doxycycline in combination with NAC can stimulate the growth of planktonic cells of *S. aureus*, which is a prerequisite for a negative outcome of therapy with both compounds. Further investigations are necessary to clarify different interactions between NAC and doxycycline against strong biofilm-forming strains of *S. aureus* and *P. aeruginosa* and against a weak biofilm-producing strain of *E. coli*. Altogether, the data from the current study indicate that NAC and doxycycline should be applied in combination with caution. However, further investigations are required to reject or to confirm the lack of interactions between doxycycline and NAC, including characterizing the biofilm with other more advanced methods. The co-administration of NAC can reduce or increase the activity of this tetracycline antibiotic against Gram-negative or Gram-positive bacteria in a strain-specific manner, either in the stage of planktonic cells or biofilm formation. NAC showed inhibitory activity towards bacteria at very high concentrations, which suggests that it can exhibit antibacterial effects after local application.

## Figures and Tables

**Figure 1 antibiotics-12-01187-f001:**
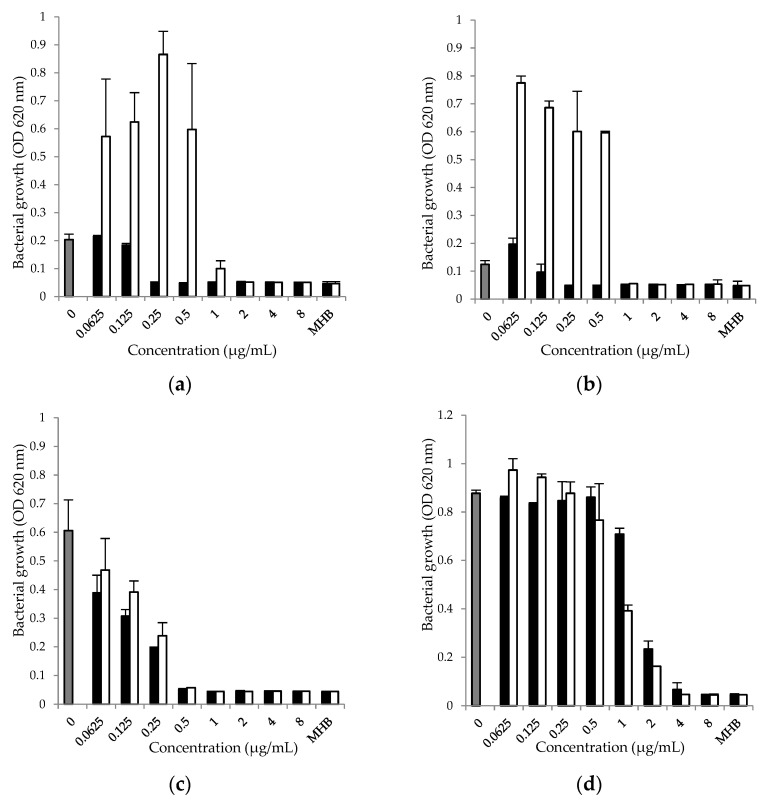
Effect of doxycycline (0.0625–8.0 μg/mL), alone (■) and in combination with N-acetyl-L-cysteine (1 μg/mL, □) on the growth of (**a**) *S. aureus* ATCC 25923; (**b**) *S. aureus* O74; (**c**) *E. coli* ATCC 25922; (**d**) *P. aeruginosa* ATCC 27853 in MHB for 24 h. The data are presented as mean ± SD of the growth rate, calculated on the basis of optical density (OD 620 nm) of the tested strain. Positive control (■)—MHB with microorganisms added and MHB—Mueller–Hinton broth without drugs and microorganisms.

**Figure 2 antibiotics-12-01187-f002:**
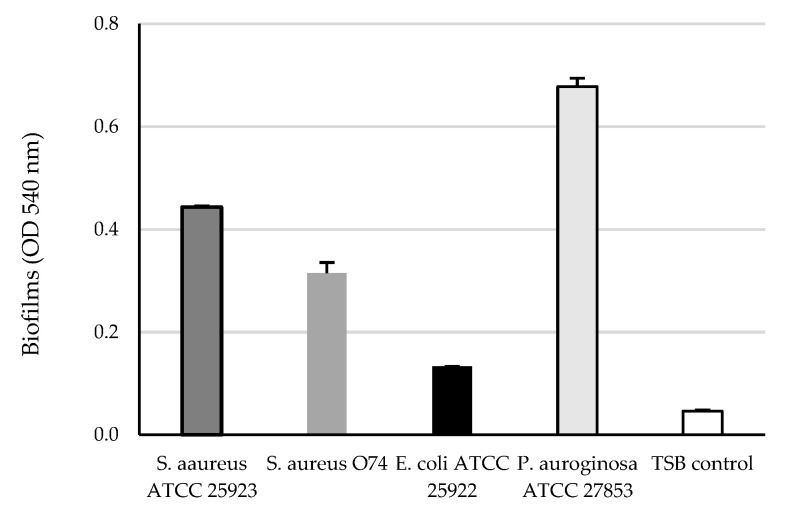
Ability of *S. aureus* ATCC 25923, *S. aureus* O74, *E. coli* ATCC 25922 and *P. aeruginosa* ATCC 27853 to form biofilm.

**Figure 3 antibiotics-12-01187-f003:**
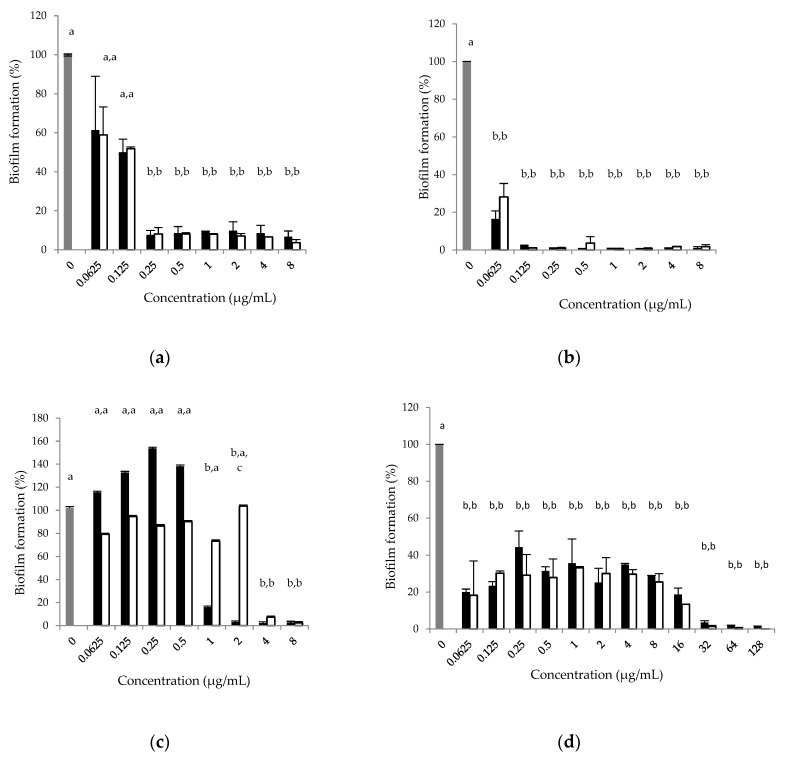
Effect of doxycycline (0.0625–128 μg/mL), alone (■) and in combination with N-acetyl-L-cysteine (1 μg/mL, □), on biofilm formation by (**a**) *S. aureus* ATCC 25923; (**b**) *S. aureus* O74; (**c**) *E. coli* ATCC 25922; and (**d**) *P. aeruginosa* ATCC 27853. The data are presented as mean ± SD of the growth rate, calculated on the basis of optical density (OD_540 nm_) of the test sample against the OD_540 nm_ of the positive control (■), expressed as percentage. a and b show the differences at *p* < 0.05 between the positive control and the test samples; c shows the differences at *p* < 0.05 between the combination of doxycycline and N-acetyl-L-cysteine and doxycycline alone.

**Table 1 antibiotics-12-01187-t001:** Minimum inhibitory concentrations (MICs) of doxycycline hyclate, N-acetyl-L-cysteine and their combination against *S. aureus* ATCC 25923, *S. aureus* O74, *E. coli* ATCC 25922 and *P. aeruginosa* ATCC 27853.

Tested Strains	MIC		
	Doxycycline	Doxycycline and N-Acetyl-L-cysteine	N-Acetyl-L-cysteine
*S. aureus* ATCC 25923	0.25 μg/mL	1 μg/mL	4000 µg/mL
*S. aureus* O74	0.25 μg/mL	1 μg/mL	4000 µg/mL
*E. coli* ATCC 25922	0.5 μg/mL	0.5 μg/mL	4000 µg/mL
*P. aeruginosa* ATCC 27853	4 μg/mL	4 μg/mL	4000 µg/mL

**Table 2 antibiotics-12-01187-t002:** Minimum biofilm inhibitory concentrations (MBICs) of doxycycline (0.0625–128 μg/mL), alone and in combination with N-acetyl-L-cysteine (1 μg/mL), and of N-acetyl-L-cysteine (0.122–8000 µg/mL) for *S. aureus* ATCC 25923, *S. aureus O74, E. coli* ATCC 25922 and *P. aeruginosa* ATCC 27853.

Tested Strains	MBIC		
	Doxycycline	Doxycycline and N-Acetyl-L-cysteine	N-Acetyl-L-cysteine
*S. aureus* ATCC 25923	0.25 μg/mL	0.25 μg/mL	4000 µg/mL
*S. aureus* O74	0.125 μg/mL	0.125 μg/mL	4000 µg/mL
*E. coli* ATCC 25922	2 μg/mL	4 μg/mL	4000 µg/mL
*P. aeruginosa* ATCC 27853	32 μg/mL	32 μg/mL	4000 µg/mL

## Data Availability

All data are contained within the article.

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
