# Peer review of "Effect of N-Acetyl-L-cysteine on Activity of Doxycycline against Biofilm-Forming Bacterial Strains"

_antibiotics, 2023, doi:10.3390/antibiotics12071187_

Round 1

Author Response

Thank you very much for the remarcs in relation to our manuscript with ID antibiotics- 2504302. We tried to answer to all remarks and to revise the manuscript according to the suggestions.

Comments and Suggestions for Authors

Reviewer 1

Answer: The remarks were listed in the attached file. We followed the suggestions and revised the text accordingly. Thank you very much for the precise indications in the text.

Lines 16-17: Is this a literature data? If so, please re-write the sentence to generalize the information.

Answer: It has been revised to: “According to previous studies N-acetyl-L-cysteine (NAC) showed inhibitory effect on biofilm formation when it was applied in combination with beta-lactam antibiotics and fluoroquinolones.”

 Please use the full names of the bacteria where they are used first.

Answer: It has been revised.

 What did the authors mean "twofold increased"? I understand that MICs that were observed against Gram-negative strains were found to be two times higher. Please re-write the sentence.

Answer: It has been revised: “The minimum inhibitory concentrations (MIC) of doxycycline were not affected by NAC for Gram-negative strains and were found to be two times higher for the strains of S. aureus.”

 Please give the obtained results here to determine the MIC and MBIC.

Answer: The results were added: “The minimum biofilm inhibitory concentrations (MBIC) for Gram-negative bacteria (2 μg/mL for E. coli ATCC 25922 and 32 μg/mL for P. aeruginosa ATCC 27853), determined by standard safranin colorimetric assay, were higher than MIC (0.5 and 4 μg/mL, respectively).”

This result is related to the antimicrobial activity of NAC alone. So, it does not directly address the purpose of the study which had been aimed to determine the synergistic effect of NAC-doxycycline. Please re-write to conclude your results about the combination of NAC-doxycycline.

Answer: The last two sentences from the abstract has been deleted and it has been revised to “NAC did not affect the strong inhibitory effect of doxycycline on biofilm formation by the strains of S. aureus.

This is not a proper scientific abbreviation for Pseudomonas aeruginosa. Please use as "P. aeruginosa" throughout the article.

Answer: It has been revised through the whole manuscript.

 Please give the full name of EMA.

Answer: It has been corrected by adding the full name.

 Please check the grmmar of this sentence and re-write.

Answer: It has been revised: “The investigation aimed to evaluate the antibacterial activity of doxycycline and NAC, used either alone or in combination, on biofilm-producing strains from Staphylococcus aureus, Pseudomonas aeruginosa and Escherichia coli.”

 The growth of bacteria is not a result. Please re-write the heading.

Answer: It has been revised: “Determination of Minimum inhibitory concentrations (MIC) of doxycycline hyclate and N-acetyl-L-cysteine”

Please interprete the relationship of Table 1 and Figure 1. I could not figure aout the relationship between two. Table 1 is the MIC of tested drugs, but why did the authors give the obtained ODs in Figrue 1?

Similar question is also valid for Table 2 and Figure 3.

 Answer: We decided to give these figures with OD values because stimulation of the growth of bacteria can be depicted with figures instead of long explanations in the text. We hope that the reviewer will accept the presentation of the data in this way. Figure 1 has been deleted from line 102, the last sentence of the paragraph. We hope that the explanations in the next paragraph are enough.

The same was done for Table 2 and Figure 3.

 Lines 288 and 290: Please give the full name of ATCC and Please use capital letters.

Answer: It has been changed.

How did the authors dissolve this amount in 1 mL of distilled water? These concentrations are too high, and already have their own optical density. So, the opasity of the bacterial suspensions may be effeced with such a high concentration. Thus, this opasity may change the calculation of MIC.

Answer: We followed the instructions of the supplier of the compound and checked the literature. The final working concentration was selected according to our previous experiments and to the information of Sigma for solubility of NAC in water (H2O: 100 mg/mL (with heating up to 37oC at a water bath). For the final experiments we decided to start with a concentration of 16 mg/ml. Indeed, we could not increase the concentration because we observed increasing of the values of OD of the solutions, containing high concentrations of NAC. I attached a figure and original data with our preliminary tests (with P. aeruginosa) and the OD of high concentrations of NAC (Please, see the attached file).

Finally, for the real experiments we decided to start with initial concentration of 16 mg/ml NAC. Thank you for this remark and we apologise for giving insufficient explanations in the text. The text has been revised in order to keep clear what was done in the current work.

This paragraph is related to the Results of the study, but not proper for Conclusion part. Please conclude your results in COnclusion part without giving the results of the study one by one.

Answer: The paragraph has been revised and the results were excluded:

 “In conclusion, our data suggest that doxycycline in combination with NAC can stimulate the growth of planktonic cells of S. aureus which is a prerequisite for negative outcome of the therapy with both compounds. Further investigations are necessary for clarification of different interactions between NAC and doxycycline against strong biofilm-forming strains of S. aureus and P. aeruginosa and against weak biofilm producing strain of E. coli. Altogether the data from the current study indicate that NAC and doxycycline should be applied in combination with caution. However, further investigations are required to reject or to confirm the lack of interactions between doxycycline and NAC, including characterization of the biofilm with other more advanced methods. Co-administration of NAC can reduce or increase the activity of this tetracycline antibiotic against Gram-negative or Gram-positive bacteria in a strain specific manner, either in the stage of planktonic cells or biofilm formation. NAC showed inhibitory activity on bacteria at very high concentrations which suggest that it can exhibit its antibacterial effect after local application.”

Reviewer 2 Report

I think that the manuscript entitled “Effect of N-Acetyl-L-cysteine on doxycycline activity against biofilm-forming bacterial strains” is, in principle, suitable for publication in the Antibiotics, Special Issue “Antibiotics Used in Animals Disease Control”. This manuscript explores the impact of N-acetyl-L-cysteine (NAC) on the effectiveness of doxycycline in combating biofilm-producing bacteria, which are often linked to persistent infections in veterinary patients. The study uses four bacterial strains (S. aureus ATCC 25923, S. aureus O74, E. coli ATCC 25922, and Ps. aeruginosa ATCC 27853) to evaluate the activity of doxycycline, both alone and in combination with NAC, on planktonic bacteria and biofilm formation. The findings suggest that the combined use of doxycycline and NAC could potentially enhance the growth of certain bacterial strains. The manuscript brings some new information to the scientific community. However, I also have minor comments:

Comments:

Line 55. "penicillin’s, tetracycline’s" should be "penicillins, tetracyclines". The apostrophe is not needed in these cases as they are plural, not possessive. Please check.
Figures 1, 3. I think, that the positive control in the descriptions of Figures #1 and #3 should be indicated as a grey square rather than a black square. Please check.
Table 2. I believe that for clarity, the concentration units for doxycycline, NAC, and their combined use in Table 2 and the entire study should consistently be given in μg/mL. The simultaneous use of μg/mL and mg/mL for MIC might lead to misunderstandings.
Lines 149-164. The paragraph starting with "NAC, administered alone, significantly inhibited the growth of S. aureus ATCC 25923 and E. coli ATCC 25922 at concentrations ≥ 2 mg/mL." is repeated twice. This is likely an error and the repeated paragraph should be removed.
Lines 197-200. "Four-fold higher values of MIC of doxycycline were registered when it was applied in combination with NAC, added at concentrations of 1 μg/mL, achievable in the body of chickens at steady-state conditions after administration at a dose rate of 100 mg/kg b.w. for five consecutive days [32]." This sentence could be more clearly written as "MIC values of doxycycline were four-fold higher when it was applied in combination with NAC, added at concentrations of 1 μg/mL. These concentrations are achievable in the bodies of chickens at steady-state conditions after administration at a dose rate of 100 mg/kg body weight for five consecutive days [32]."

In my opinion, the designation "P. aeruginosa" is more commonly used compared to "Ps. aeruginosa" for Pseudomonas aeruginosa.

Author Response

Reviewer 2

I think that the manuscript entitled “Effect of N-Acetyl-L-cysteine on doxycycline activity against biofilm-forming bacterial strains” is, in principle, suitable for publication in the Antibiotics, Special Issue“Antibiotics Used in Animals Disease Control”. This manuscript explores the impact of N-acetyl-L-cysteine (NAC) on the effectiveness of doxycycline in combating biofilm-producing bacteria, which are often linked to persistent infections in veterinary patients. The study uses four bacterial strains (S. aureus ATCC 25923, S. aureus O74, E. coli ATCC 25922, and Ps. Aeruginosa ATCC 27853) to evaluate the activity of doxycycline, both alone and in combination with NAC, on planktonic bacteria and biofilm formation. The findings suggest that the combined use of doxycycline and NAC could potentially enhance the growth of certain bacterial strains. The manuscript brings some new information to the scientific community. However, I also have minor comments:

We are thankful for the opinion of the Reviewer and for the suggestions.

Comments:

Line 55. "penicillin’s, tetracycline’s" should be "penicillins, tetracyclines". The apostrophe is not needed in these cases as they are plural, not possessive. Please check.

Answer: The names were revised, thank you for this remark.

Figures 1, 3. I think, that the positive control in the descriptions of Figures #1 and #3 should be indicated as a grey square rather than a black square. Please check.

Answer: It has been corrected.

Table 2. I believe that for clarity, the concentration units for doxycycline, NAC, and their combined use in Table 2 and the entire study should consistently be given in μg/mL. The simultaneous use of μg/mL and mg/mL for MIC might lead to misunderstandings.

Answer: It has been corrected in the entire manuscript and the data were presented in μg/mL.

Lines 149-164. The paragraph starting with "NAC, administered alone, significantly inhibited the growth of S. aureus ATCC 25923 and E. coli ATCC 25922 at concentrations ≥ 2 mg/mL." is repeated twice. This is likely an error and the repeated paragraph should be removed.

Answer: Thank you for this remark. The repetition has been deleeeted.

 Lines 197-200. "Four-fold higher values of MIC of doxycycline were registered when it was applied in combination with NAC, added at concentrations of 1 μg/mL, achievable in the body of chickens at steady-state conditions after administration at a dose rate of 100 mg/kg b.w. for five consecutive days [32]."This sentence could be more clearly written as "MIC values of doxycycline were four-fold higher when it was applied in combination with NAC, added at concentrations of 1 μg/mL. These concentrations are achievable in the bodies of chickens at steady-state conditions after administration at a dose rate of 100 mg/kg body weight for five consecutive days [32]."

Answer: Thank you for your suggestion. The sentence has been revised.

In my opinion, the designation "P. aeruginosa" is more commonly used compared to "Ps. aeruginosa" for Pseudomonas aeruginosa.

Answer: It has been revised through the entire manuscript.

We also used English editing survices and I attached the certificate.
